# The m^7^G Reader NCBP2 Promotes Pancreatic Cancer Progression by Upregulating MAPK/ERK Signaling

**DOI:** 10.3390/cancers15225454

**Published:** 2023-11-17

**Authors:** Jiancong Xie, Taiwei Mo, Ruibing Li, Hao Zhang, Guanzhan Liang, Tao Ma, Jing Chen, Hanlin Xie, Xiaofeng Wen, Tuo Hu, Zhenyu Xian, Weidong Pan

**Affiliations:** 1Department of General Surgery (Pancreatic Hepatobiliary Surgery), The Sixth Affiliated Hospital, Sun Yat-sen University, Guangzhou 510655, China; xiejc6@mail2.sysu.edu.cn (J.X.); zhangh677@mail.sysu.edu.cn (H.Z.); matao7@mail2.sysu.edu.cn (T.M.); 2Department of General Surgery, The First Affiliated Hospital of Jinan University, Guangzhou 510630, China; motaiweiray@163.com; 3Department of General Surgery (Colorectal Surgery), The Sixth Affiliated Hospital, Sun Yat-sen University, Guangzhou 510655, China; lirb6@mail2.sysu.edu.cn (R.L.); lianggzh8@mail2.sysu.edu.cn (G.L.); chenj396@mail2.sysu.edu.cn (J.C.); xiehlin3@mail2.sysu.edu.cn (H.X.); wenxf6@mail.sysu.edu.cn (X.W.); hutuo3@mail.sysu.edu.cn (T.H.); 4Guangdong Provincial Key Laboratory of Colorectal and Pelvic Floor Diseases, Guangdong Institute of Gastroenterology, The Sixth Affiliated Hospital, Sun Yat-sen University, Guangzhou 510655, China; 5Biomedical Innovation Center, The Sixth Affiliated Hospital, Sun Yat-sen University, Guangzhou 510655, China

**Keywords:** m^7^G methylation, NCBP2, MEK/ERK, c-JUN, pancreatic adenocarcinoma

## Abstract

**Simple Summary:**

N7-methylguanosine (m^7^G) methylation plays an important role in the development of pancreatic adenocarcinoma (PDAC). Among the 29 regulatory genes of m^7^G methylation, they can be roughly divided into three categories: methyltransferase (writer), demethylase (eraser), and binding protein (reader) according to their functions. In this study, we explore the role of m^7^G reader NCBP2 in PDAC progression: NCBP2 activates MEK/ERK signaling by upregulating c-JUN, thereby promoting PDAC cell proliferation. In summary, we identify a novel potential therapeutic target for PDAC, which sheds new light on the treatment of PDAC patients.

**Abstract:**

PDAC is one of the most common malignant tumors worldwide. The difficulty of early diagnosis and lack of effective treatment are the main reasons for its poor prognosis. Therefore, it is urgent to identify novel diagnostic and therapeutic targets for PDAC patients. The m^7^G methylation is a common type of RNA modification that plays a pivotal role in regulating tumor development. However, the correlation between m^7^G regulatory genes and PDAC progression remains unclear. By integrating gene expression and related clinical information of PDAC patients from TCGA and GEO cohorts, m^7^G binding protein NCBP2 was found to be highly expressed in PDAC patients. More importantly, PDAC patients with high NCBP2 expression had a worse prognosis. Stable NCBP2-knockdown and overexpression PDAC cell lines were constructed to further perform in-vitro and in-vivo experiments. NCBP2-knockdown significantly inhibited PDAC cell proliferation, while overexpression of NCBP2 dramatically promoted PDAC cell growth. Mechanistically, NCBP2 enhanced the translation of c-JUN, which in turn activated MEK/ERK signaling to promote PDAC progression. In conclusion, our study reveals that m^7^G reader NCBP2 promotes PDAC progression by activating MEK/ERK pathway, which could serve as a novel therapeutic target for PDAC patients.

## 1. Introduction

The global burden of PDAC has dramatically increased over the past few decades, which is expected to be a leading cause of cancer-related mortality [1]. The extensive fibrosis within PDAC tissues, termed as desmoplasia, dramatically reduces the delivery and efficacy of chemotherapy by decreasing vessel perfusion [2]. In addition, major histocompatibility complex I (MHC I) expression is downregulated in PDAC, which prevents the recognition of PDAC cells by CD8^+^ T cells and promotes their immune evasion [3]. Although the 5-year overall survival rate of PDAC patients has improved from 3% to around 9% in recent years, it is extremely low compared with other cancer types [4]. In most PDAC cases, the low survival rate is due to the advanced stage at diagnosis and lack of effective treatment. Therefore, there is an urgent need to identify novel therapeutic targets for PDAC to improve their prognosis.

There are more than 100 different modification types identified in RNA [5,6,7,8]. Among them, m^7^G methylation is one of the most common RNA modification types [9,10,11]. The m^7^G methylation regulates the function of tRNA, rRNA, and mRNA, which plays a crucial role in various pathological processes, including cancer progression, autoimmune disorders, and cardiac injury repair [12,13,14,15]. Ming Kuang et al. demonstrated that m^7^G methyltransferase complex protein methyltransferase-like 1 (METTL1) and WD repeat domain 4 (WDR4) promoted Lenvatinib resistance in hepatocellular carcinoma by increasing the translation of EGFR pathway genes [16]. In addition, METTL1 and WDR4 accelerate the progression of esophageal squamous cell carcinoma by enhancing the translation of oncogenic transcripts from RPTOR/ULK1/autophagy pathway [17]. However, the role of m^7^G regulators in PDAC progression is still unclear.

The nuclear cap-binding protein (NCBP) family members include NCBP1, NCBP2, and NCBP3 [18]. Among these, NCBP2 and NCBP1 form the cap-binding complex (CBC), which serves as the central factor to orchestrate downstream RNA biogenesis [19,20,21,22]. Besides stabilizing NCBP2, NCBP1 can serve as an adaptor for other RNA processing factors in CBC, while NCBP2 directly interacts with the RNA cap and then enables CBC to bind with RNA [18]. Like NCBP2, NCBP3 can substitute NCBP2 to bind with NCBP1 and form an alternative CBC in some special circumstances [18]. In addition, previous research has demonstrated that NCBP3 could interact with the exon junction complex as well as the transcription and export complex to promote mRNA expression [23]. Moreover, the CBC can coordinate most downstream RNA biogenesis processes such as 3′-end processing, pre-mRNA splicing, nuclear-cytoplasmic transport, nonsense-mediated decay, and recruitment of translation factors in the cytoplasm [21,22,24,25,26]. Therefore, NCBP family members play an important role in RNA modification. However, their role as m^7^G regulators in PDAC is still obscure.

Mitogen-activated protein kinase (MAPK) is an important transmitter of signals from the cell surface to the interior of the nucleus. Its family members include MAPK kinase kinase (MAPKKK), MAPK kinase (MAPKK), and MAPK [27]. There are four MAPK cascades, including ERK1/2, p38 MAPK, c-JUN N-terminal kinase (JNK), and ERK5 [27]. Studies have reported that RAF/MEK/ERK signaling pathway is a fundamental pathway in cell biology [28]. RAF/MEK/ERK pathway not only participates in the regulation of important physiological processes, but its dysregulation also leads to the development of various human diseases, including cancer [28,29]. The activating mutations and dysfunction of RAF/MEK/ERK signaling pathway are prevalent in most cancer types [30,31]. However, the precise regulatory mechanism for the hyperactivation of MEK/ERK signaling in PDAC remains unclear.

In this study, we find that m^7^G reader NCBP2 is highly expressed and associated with poor prognosis in PDAC patients by integrating RNA-seq and DNA microarray data from public databases. Mechanistically, NCBP2 increased the translation of c-JUN to activate MEK/ERK pathway, thereby promoting PDAC progression. Therefore, our study identifies that NCBP2, a pivotal regulator for PDAC progression, could serve as a novel therapeutic target for PDAC patients.

## 2. Materials and Methods

### 2.1. Bioinformatics Analysis

In this study, RNA-seq, DNA microarray data, and related clinical data were obtained from TCGA-PDAC and GEO datasets (GSE15471, GSE28735, GSE16515). Gene set enrichment analysis (GSEA) software (GSEA 4.3.2, Broad Institute, Boston, MA, USA) was used for gene enrichment analysis. The URLs of all online analysis websites are shown in Appendix A.

### 2.2. Cell Culture

We obtained human embryonic kidney epithelial cell line 293T (HEK293T) and PDAC cell lines (Panc 05.04, PANC-1, AsPC-1, BxPC-3, MIA PaCa-2) from the American Type Culture Collection (ATCC, Manassas, VA, USA). These cells were cultured in DMEM (Corning, 10-013-CVRC, Corning, NY, USA) or RPMI 1640 (Corning, 10-040-CVRC, NY, USA) supplemented with 10% fetal bovine serum (Sigma Aldrich, F8318, St. Louis, MO, USA) and 1% penicillin-streptomycin (Corning, 30-002-CI, NY, USA). The human pancreatic duct epithelial cell line HPDE/E6E7 (referred to as HPDE) was purchased from Kerafast (Boston, MA, USA) [32]. HPDE were routinely cultured in keratinocyte serum-free (KSF) medium supplemented with epidermal growth factor (Gibco-BRL, Grand Island, NY, USA) and bovine pituitary extract. The cells were cultured in a humidified incubator with 5% CO_2_ at 37 °C. We performed mycoplasma contamination testing on all cell lines used in this study every four months (last test date: 26 May 2023).

### 2.3. Immunohistochemistry

The expression of Ki67 in PDAC tissues from nude mice and NCBP2, c-JUN, p-ERK in a tissue microarray (Shanghai Outdo Biotech Co., Ltd., HPanA120PG01-M, Shanghai, China) were evaluated by immunohistochemical (IHC) staining. For IHC analysis, paraffin sections were deparaffinized in xylene and rehydrated by graded alcohols. Then, the slides were microwaved in citrate antigen retrieval solution (Servicebio, G1202, Wuhan, China) and incubated in 3% hydrogen peroxide solution for 10 min. After blocking with 5% BSA (Servicebio, GC305010, Wuhan, China) for 20 min, primary antibody solution was added to these slides and incubated at 4 °C overnight. Each slide was probed with 100 μL horseradish peroxidase (HRP)-labeled secondary antibody for 60 min. The slides were incubated with 100 μL diaminobenzidine (DAB) solution (Servicebio, G1215-200T, Wuhan, China) and then stained with hematoxylin (Servicebio, G1004, Wuhan, China).

### 2.4. RNA Extraction and Real-Time Quantitative PCR (RT-qPCR)

The TRIzol Reagent (Sigma Aldrich, T9424, St. Louis, MO, USA) was used to extract total RNA, and ReverTra Ace qPCR RT Kit (Toyobo, FSQ-101, Osaka, Japan) was applied for the reverse transcription reaction. RT-qPCR detection was then performed using 2 × S6 Universal SYBR qPCR Mix (Thermo Fisher, Q204-01, Waltham, MA, USA) and ABI QuantStudio™ 7 Flex Real-Time PCR Systems. The expression level of indicated genes was normalized to 18s rRNA using the 2^−ΔΔCt^ method. All PCR primers used in this study are listed in Appendix A.

### 2.5. Western Blotting

Total cellular protein was extracted using RIPA buffer (Beyotime, P0013E, Shanghai, China). After electrophoresis, transferring, and blocking, the corresponding diluted primary antibody was added and incubated overnight at 4 °C. The next day, PVDF membrane was incubated with the indicated secondary antibody. The antibodies used are listed in Appendix A. Bio-Rad (Hercules, CA, USA) chemiluminescence imaging system was used to detect the luminescent signals.

### 2.6. Plasmids Construction and siRNA Transfection

The coding sequence of NCBP2 and c-JUN gene was cloned into pSin-vector (Addgene, Watertown, MA, USA, #16577) to generate NCBP2-overexpression and c-JUN-overexpression plasmid. The shRNA sequence of NCBP2 was cloned into pLKO.1-puro vector (Addgene, Watertown, MA, USA, #8453) to construct shRNA plasmid targeting NCBP2. The plasmids, together with package plasmid pMD2.G (Addgene, Watertown, MA, USA, #12259) and psPAX2 (Addgene, Watertown, MA, USA, #12260), were transfected into HEK293T cells using TurboFect (Thermo Fisher, R0531, Waltham, MA, USA), then lentivirus was collected 24 and 48 h after transfection. Stable NCBP2-knockdown, NCBP2-overexpression, and c-JUN-overexpression PDAC cell lines were constructed by infecting PDAC cells using the lentivirus with polybrene (1:1000) and then selected with puromycin. SignaGen GenMute™ siRNA transfection reagent (BIOFIVEN, SL100568, Guangzhou, China) was used to transfect c-JUN-targeting siRNA into PDAC cells. The siRNA sequence and primers for NCBP2-targeting shRNA, as well as NCBP2-overexpression and c-JUN-overexpression plasmid construction, are listed in Appendix A.

### 2.7. Nude Mice Xenograft Tumor Model

Five-week-old BALB/c nude mice were ordered from GemPharmatech (San Diego, CA, USA). All nude mice were raised and monitored under specific pathogen-free (SPF) conditions at the animal facility of Guangdong Pharmaceutical University. The control and NCBP2-knockdown Panc 05.04 cells were orthotopically inoculated into the pancreas of BALB/c nude mice. When the orthotopic pancreatic tumor grew to the expected size, the mice were sacrificed, and then, tumor tissues were collected and weighed. The animal experiment was approved by the Ethics Committee of Guangdong Pharmaceutical University (Ethics No. gdpulac2023253).

### 2.8. Cell Proliferation Assays

For the colony formation assay, Panc 05.04, PANC-1, AsPC-1 and BxPC-3 cells from indicated groups were seeded in 6-well plates (500 cells/per well). Cells were grown in DMEM or RPMI 1640 supplemented with 10% fetal bovine serum and 1% penicillin-streptomycin, the medium was changed every 3 days. Two weeks later, colonies were fixed with 4% paraformaldehyde (4% PFA), followed by staining with 0.1% crystal violet (MedChemExpress, HY-B0324A, Monmouth Junction, NJ, USA) for 30 min and washing with pure water. The number of colonies was counted.

For cell growth curve assay, Panc 05.04, PANC-1, AsPC-1, and BxPC-3 cells from indicated groups were seeded in 6-well plates (20,000 cells/per well for Panc 05.04, PANC-1; 30,000 cells/per well for BxPC-3; 40,000 cells/per well for AsPC-1). Cells were grown in DMEM or RPMI 1640 supplemented with 10% fetal bovine serum and 1% penicillin-streptomycin, cell culture medium was replaced every 3 days. Cells were digested with trypsin and stained with trypan blue, then alive cells were counted using a hemocytometer.

For 3D sphere formation assay, Panc 05.04, PANC-1, AsPC-1, and BxPC-3 cells from indicated groups were seeded in the ultra-low absorption 96-well plates (1000 cells/per well; LV-Biotech, LV-ULA002-96UW, Shenzhen, China). Cells were cultured in DMEM or RPMI-1640 containing 10% fetal bovine serum and 1% penicillin-streptomycin. The cell culture medium was refreshed every 3 days. Images were captured using a microscope after 14 days.

### 2.9. Methylated RNA Immunoprecipitation (Me-RIP) Assay

The m^7^G site on c-JUN mRNA in PDAC cells was investigated by Me-RIP assay. A fragmentation buffer was added to RNA samples, and then, RNA samples were fragmented into approximately 100 nucleotides, which were finally purified by RNase MiniElute kit. Specific enrichment was performed using m^7^G antibody (BioVision, San Francisco, CA, USA) conjugated with Protein A/G Magnetic Beads (MedChemExpress, MCE HY-K0202-5 mL, Monmouth Junction, NJ, USA). After elution, reverse transcription, and amplification, qPCR was performed on RNA input, IgG-IP fragment, and m^7^G-IP fragment samples. The signal intensity was calculated based on the corresponding CT values.

### 2.10. Polysome Profiling Analysis

PDAC cells were cultured in the 15 cm plate and treated with 100 μg/mL cycloheximide (Sigma-Aldrich, C7698, St. Louis, MO, USA) for 10 min. After washing with cold DPBS (Thermo Fisher, 14190144, Waltham, MA, USA), the cells were then lysed by lysis buffer. To purify lysate, the lysed cells were centrifuged at 11,200× *g* for 8 min under 4 °C. After that, 1 mL supernatant was loaded on the top of SW41 ultracentrifuge tubes (Backman, 331372, Milwaukee, WI, USA) containing 10–50% (*w*/*v*) sucrose density gradients (BioComp Instruments, Fredericton, NB, Canada). Then, the tubes were centrifuged at 260,000× *g* for 2 h using SW41 rotor under 4 °C. Finally, gradients were fractionated and monitored at an absorbance of 254 nm (VICTOR Nivo, Waltham, MA, USA).

### 2.11. RNA Immunoprecipitation

PDAC cells were washed with PBS and then lysed by IP lysis buffer. After incubation for 30 min, the lysate was centrifuged at 12,000× *g* for 10 min at 4 °C. The Protein A/G Magnetic Beads (MedChemExpress, MCE HY-K0202-5 mL, Monmouth Junction, NJ, USA) and antibodies (IgG or NCBP2 antibody) were added into the lysate and incubated overnight at 4 °C. After washing with wash buffer 3 times, co-precipitated RNAs were extracted by TRIzol Reagent. After reverse transcription and amplification, qPCR was performed on RNA input, IgG-IP fragment, and NCBP2-IP fragment samples. The signal intensity was calculated based on the corresponding CT values.

### 2.12. Statistical Analysis

All statistical analyses were performed using Prism 7.0 (GraphPad Software, La Jolla, CA, USA). Student’s *t*-test was utilized for the comparison between the two groups, and the Kaplan–Meier method was used for survival analysis. The data from the cell proliferation assay, 3D sphere, and colony formation assay were analyzed by one-way analysis of variance (ANOVA). The correlation between indicated genes was analyzed by Pearson correlation analysis. And *p* < 0.05 was considered as statistically significant.

## 3. Results

### 3.1. NCBP2 Is Highly Expressed in PDAC and Associated with Poor Prognosis

A total of 29 m^7^G-related genes were analyzed using GEPIA 2.0 and GEO cohorts (Figure 1). NCBP2, AGO2, LSM1, EIF4E2, and EIF4G3 were significantly differentially expressed in PDAC and normal pancreatic tissues (Figure 2A,B and Appendix A). The Kaplan–Meier survival analysis of these five genes was performed by GEPIA 2.0, and only NCBP2 was significantly associated with the prognosis of PDAC patients (Figure 2C,D and Appendix A). Furthermore, the subgroup analysis results indicated that the association between survival and NCBP2 expression was much more significant in stage I-IIA PDAC patients compared with that in stage IIB-IV patients (Appendix A). Similarly, the correlation between survival and NCBP2 expression was much more remarkable in grade 1–2 PDAC patients compared with that in grade 3–4 patients (Appendix A). These data indicate that NCBP2 is highly expressed in PDAC tissues and related with poor prognosis, especially in early-stage patients.

To further investigate the role of NCBP2 in regulating PDAC progression, we evaluated the mRNA and protein levels of NCBP2 in a normal pancreatic cell line (HPDE) and five PDAC cell lines (Panc 05.04, PANC-1, MIA PaCa-2, BxPC-3, AsPC-1). The expression of NCBP2 was relatively higher in Panc 05.04 and PANC-1 cells, while MIA PaCa-2, BxPC-3, and AsPC-1 cells were exhibited with relatively low NCBP2 expression (Figure 2E,F). In addition, we detected NCBP2 expression level in the tissue microarray of PDAC patients. The expression level of NCBP2 in PDAC was significantly higher than that in para-carcinoma tissue (Figure 2G). Taken together, these results suggest that NCBP2 may be a promising biomarker for the diagnosis and prognostic prediction of PDAC patients.

### 3.2. NCBP2 Promotes PDAC Cell Growth In Vitro and In Vivo

To further investigate the biological significance of NCBP2 in PDAC, we designed shRNA to knockdown NCBP2 in Panc 05.04 and PANC-1 cells with relatively high NCBP2 expression (Appendix A). Knockdown of NCBP2 significantly inhibited PDAC cell proliferation, as evidenced by cell counting assay (Figure 3A). Moreover, the cell growth inhibition effect was also observed in NCBP2-knockdown PDAC cells, as shown by 3D sphere assay and colony formation results (Figure 3B,C). To further confirm the oncogenic role of NCBP2 in PDAC, we overexpressed NCBP2 in AsPC-1 and BxPC-3 cells with relatively low NCBP2 expression (Appendix A). As expected, NCBP2-overexpression dramatically promoted PDAC cell growth, as indicated by cell counting, 3D sphere assay, and colony formation assays (Figure 3D–F). These results demonstrate that NCBP2 remarkably promotes PDAC cell proliferation in-vitro.

Subsequently, we evaluated the effect of NCBP2 on pancreatic tumor growth in-vivo. The orthotopic pancreatic tumor model was constructed by injecting control and NCBP2-knockdown Panc 05.04 cells into the pancreas of nude mice. The tumor growth was significantly slower in nude mice implanted with NCBP2-knockdown Panc 05.04 cells (Figure 4A–C). Consistently, immunohistochemical results revealed that the expression of Ki-67 was remarkably decreased in NCBP2-knockdown tumor tissues compared with the control group (Figure 4D,E). Moreover, there was no significant difference in body weight between these two groups of mice (Figure 4F). Taken together, our results suggest that NCBP2 significantly promotes PDAC cell growth both in-vitro and in-vivo.

### 3.3. NCBP2 Activates MEK/ERK Signaling Pathway via c-JUN

To elucidate the underlying molecular mechanism by which NCBP2 regulated PDAC cell growth, we performed gene set enrichment analysis (GSEA) using RNA-seq data from TCGA-PDAC cohort. NCBP2 was significantly associated with MAPK signaling pathway (Figure 5A,B). To test our hypothesis, we performed Western blot analysis in control and NCBP2-knockdown PDAC cell lines. We found that MEK/ERK signaling pathway was significantly inhibited in NCBP2-knockdown PDAC cells (Figure 5C). Moreover, the MEK/ERK signaling pathway was significantly activated when NCBP2 was overexpressed (Appendix A). It has been reported that cap-binding complex (CBC) functions mainly by regulating mRNA translation efficiency [33,34,35]. Moreover, the m^7^G reader NCBP2 could also function as a CBC in an m^7^G methylation-dependent manner [19,23,36]. Therefore, we hypothesized that NCBP2 did not activate MEK/ERK signaling directly, and NCBP2 might upregulate MEK/ERK signaling via an intermediate molecule by regulating its mRNA translation. To verify our assumption, we reviewed relevant literature and found that c-JUN could directly activate the MEK/ERK signaling pathway. Knockdown of c-JUN dramatically inhibits the phosphorylation level of MEK and ERK [37,38]. In line with these, knockdown of c-JUN significantly inhibited the activation of MEK/ERK signaling in PDAC cells (Figure 5D). To verify the role of c-JUN in mediating NCBP2-activated MEK/ERK signaling, we evaluated c-JUN expression level in control and NCBP2-knockdown PDAC cells. As expected, knockdown of NCBP2 remarkably reduced the protein level of c-JUN without affecting its mRNA level in PDAC cells (Figure 5E,F). These data preliminarily demonstrate that NCBP2 activates MEK/ERK signaling via c-JUN.

As the protein level but not mRNA level of c-JUN was significantly decreased upon NCBP2 knockdown, we hypothesized that NCBP2 might regulate the protein stability or translation efficiency of c-JUN. Then, we treated control and NCBP2-knockdown PDAC cells with protein synthesis inhibitor cycloheximide (CHX). Western blot analysis results indicated that knockdown of NCBP2 had no effect on the protein stability of c-JUN in Panc 05.04 and PANC-1 cells (Appendix A). To further test our hypothesis, we conducted polysome profiling and measured the level of polysome in control and NCBP2-knockdown PDAC cells. As expected, NCBP2-Knockdown remarkably inhibited the global activation of translation (Figure 5G). And then, we assessed the m^7^G modification status of c-JUN mRNA by gene-specific m^7^G experiments, and a significant enrichment of c-JUN mRNA with specific m^7^G antibodies was observed (Figure 5H). In addition, RIP-qPCR assay result further confirmed the interaction between NCBP2 and c-JUN mRNA in PDAC cells (Appendix A). Altogether, c-JUN is a downstream target gene of NCBP2, and NCBP2 activates MEK/ERK signaling by increasing the translation efficiency of c-JUN in an m^7^G methylation-dependent manner.

### 3.4. NCBP2 Promotes PDAC Progression by Activating c-JUN/MEK/ERK Pathway

The above-mentioned results identified that c-JUN mediated the activation of MEK/ERK signaling induced by NCBP2. We then investigated the role of c-JUN in NCBP2-regulated PDAC cell proliferation. As shown by the immunoblotting results, c-JUN overexpression rescued the inhibitory effect of NCBP2-knockdown on MEK/ERK signaling (Figure 6A). Moreover, overexpression of c-JUN dramatically rescued the suppressive effect of NCBP2-knockdown on PDAC cell proliferation as evidenced by 3D sphere, cell counting, and colony formation assay results (Figure 6C and Appendix A). In addition, knockdown of c-JUN significantly inhibited the activation of MEK/ERK signaling mediated by NCBP2-overexpression (Figure 6B). More importantly, knockdown of c-JUN remarkably diminished the proliferation promotion effect mediated by NCBP2-overexpression on PDAC cells (Figure 6D and Appendix A).

To further test our conclusion, we compared the expression level of NCBP2, c-JUN, and p-ERK between PDAC and para-carcinoma tissue by IHC staining on tissue microarray. As expected, the expression levels of NCBP2, c-JUN, and p-ERK were significantly higher in PDAC tissues than those in para-carcinoma tissues (Figure 7A,B). The correlation analysis results indicated that the expression of NCBP2 was positively correlated with c-JUN and p-ERK in PDAC tissues. Meanwhile, the expression level of c-JUN was found to be positively correlated with p-ERK in PDAC tissues (Figure 7C). In summary, NCBP2 promotes PDAC cell proliferation by upregulating c-JUN-mediated MEK/ERK signaling activation (Figure 7D).

## 4. Discussion

In our study, we reveal that NCBP2 is significantly upregulated in PDAC and associated with poor prognosis in PDAC patients. Although there are several reports revealing the high expression of NCBP2 in different cancer types, its specific function and related molecular mechanisms in cancer development are still unclear [39,40]. Here, we reveal that NCBP2 increases the translation of c-JUN in an m^7^G-dependent manner to activate MEK/ERK signaling, thereby promoting PDAC progression. Therefore, NCBP2 serves as an oncogene which accelerates PDAC evolution. Our findings enable us to have a better understanding of the regulatory function of RNA m^7^G methylation on PDAC progression.

The RNA m^7^G methylation plays a vital role in the initiation and development of various cancers [41,42,43,44]. In mammals, the most studied m^7^G regulatory factor is METTL1, which binds with cofactor WDR4 to mediate m^7^G modifications in tRNA, miRNA, and mRNA [45]. The expression levels of METTL1 and WDR4 are significantly increased in human cancers and correlated with poor prognosis [12,17,46,47,48]. The main m^7^G readers are CBC and eIF4E [22]. Many studies have shown that eIF4E is highly expressed in various cancer types (breast cancer, colon cancer, bladder cancer, etc.), which enhances their metastasis and angiogenesis [49,50,51,52,53,54,55,56,57]. In breast cancer, eIF4E promotes the expression of FGF-2 by regulating its translation efficiency, thereby promoting the tumorigenicity and angiogenesis of breast cancer [49]. As a CBC, NCBP2 could recognize the m^7^G “cap” on RNA and promote RNA maturation, transportation, and translation in conjunction with eIF4E [18,33,34,35,58,59,60,61,62]. Moreover, several studies have indicated that NCBP2 promotes the proliferation, metastasis, and immune escape of oral squamous cell carcinoma and head and neck squamous cell carcinoma [63,64]. Consistent with these findings, we identified that NCBP2 promotes PDAC progression by upregulating c-JUN-mediated MEK/ERK signaling activation in an m^7^G methylation-dependent manner. Therefore, our findings further expand the understanding of NCBP2 in regulating cancer progression.

Among all MAPKKK-MAPKK-MAPK signaling pathways, the RAF-MEK-ERK pathway is widely studied [28]. The RAF-MEK-ERK axis not only participates in the regulation of physiological processes in normal cells, such as proliferation, differentiation, cell cycle, apoptosis, etc. [27,28,29]; it also plays a vital role in regulating the initiation and development of various cancers [27,65]. The hyperactivated RAF-MEK-ERK signaling promotes tumor proliferation, metastasis, extracellular matrix degradation, angiogenesis, and therapy resistance [27,65,66,67]. The c-JUN is a well-studied protein involved in various physiological and pathological processes, such as cell proliferation, apoptosis, tumorigenesis, and tissue morphogenesis [68,69,70]. Daimin Xiang et al. have demonstrated that c-JUN promotes cancer progression by directly increasing the phosphorylation level MEK/ERK signaling [37]. In line with this, our research reveals that NCBP2 recognizes m^7^G methylated-c-JUN and increases its translation efficiency, thereby activating MEK/ERK signaling. Here we uncover a novel regulatory mechanism for c-JUN expression, that is, NCBP2 promotes the translation of c-JUN mRNA to increase its protein expression in an m^7^G methylation-dependent manner. This finding may provide new ideas for the study of regulatory mechanisms in c-JUN expression.

MEK inhibitors have been well studied in hyperactivated MAPK pathway-driven tumors. Some MEK inhibitors have been approved by the FDA and applied in clinical use [71,72]. Although monotherapy with MEK inhibitors has shown limited clinical efficacy in patients with KRAS-mutant tumors, in other types of RAS mutant cancer patients, MEK inhibitors alone have exhibited considerable antitumor activity [73,74,75,76,77,78]. In addition, the combination of MEK inhibitor with BRAF or BCL-XL inhibitors have shown better clinical outcomes in cancer patients than a single drug [79,80,81,82]. Although many different ERK inhibitors have been developed, there are still relatively few ERK inhibitors available for clinical use compared with MEK inhibitors. TIC10 (Dual AKT and ERK inhibitor) is the only one approved by FDA for the treatment of H3K27m mutant glioma, which is under phase 3 clinical trial [83]. Our research results show that overexpression of NCBP2 activates the MEK/ERK signaling and thereby promotes PDAC progression. Based on these findings, we speculate that for PDAC patients with amplification or high expression of NCBP2 gene, MEK or ERK inhibitors may serve as a potential therapeutic strategy, which needs further studies.

## 5. Conclusions

In summary, our study identifies that m^7^G reader NCBP2 is upregulated and associated with poor prognosis of PDAC patients. NCBP2 promotes PDAC progression by activating MEK/ERK signaling via regulating c-JUN translation in an m^7^G-dependent manner. Therefore, NCBP2 may serve as a promising therapeutic target for PDAC patients.

## Figures and Tables

**Figure 1 cancers-15-05454-f001:**
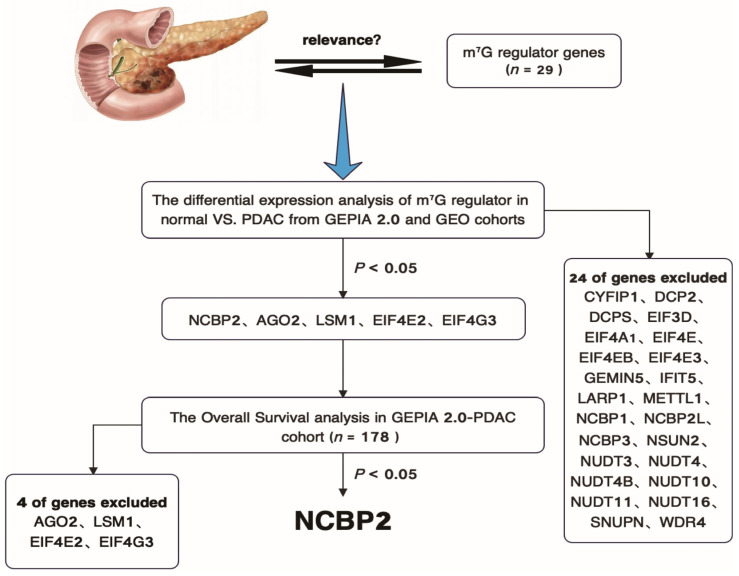
Flow chart of target m^7^G regulator gene screening process in PDAC.

**Figure 2 cancers-15-05454-f002:**
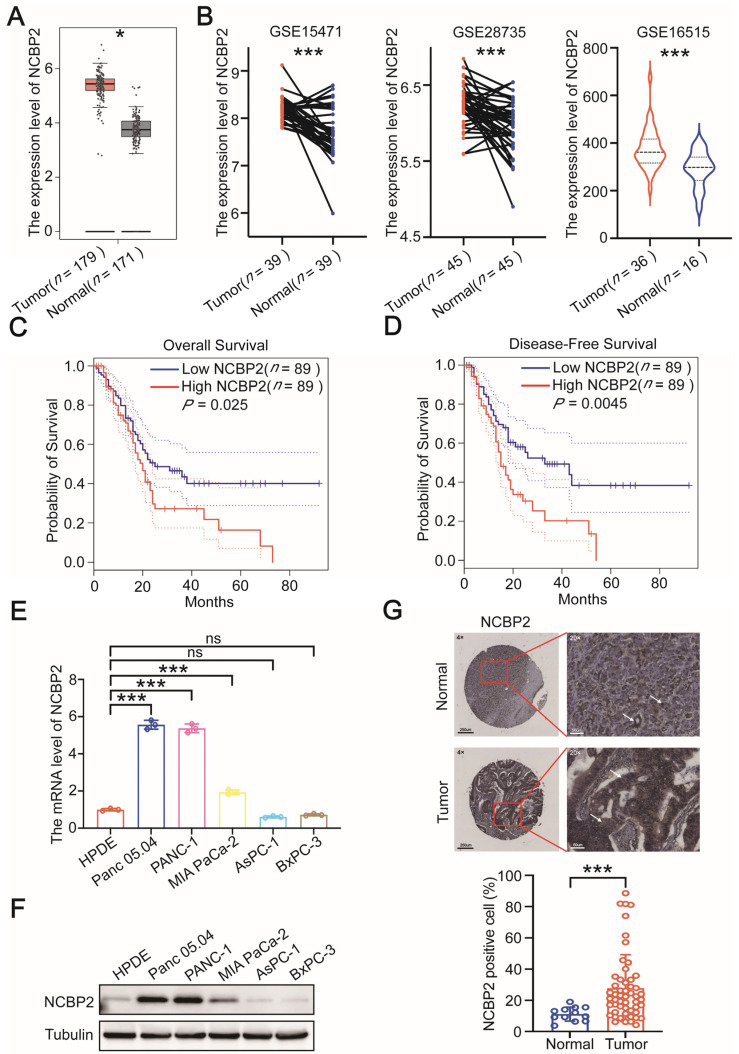
NCBP2 is significantly expressed in PDAC patients and correlated with poor prognosis. (**A**) Analysis of NCBP2 transcript levels in PDAC and normal tissues based on the RNA-seq data from GEPIA 2.0 cohort. (**B**) Analysis of NCBP2 transcript levels in PDAC and normal tissues based on DNA microarray data from GEO cohorts (GSE15471, GSE28735, GSE16515). (**C**,**D**) Overall survival and disease-free survival curves for PDAC patients with high or low NCBP2 expression in GEPIA 2.0 cohort. (**E**,**F**) The RNA and protein expression level of NCBP2 in HPDE and five PDAC cell lines. (**G**) The expression level of NCBP2 in PDAC and para-carcinoma tissues from tissue microarray, * *p* < 0.05 and *** *p* < 0.001.

**Figure 3 cancers-15-05454-f003:**
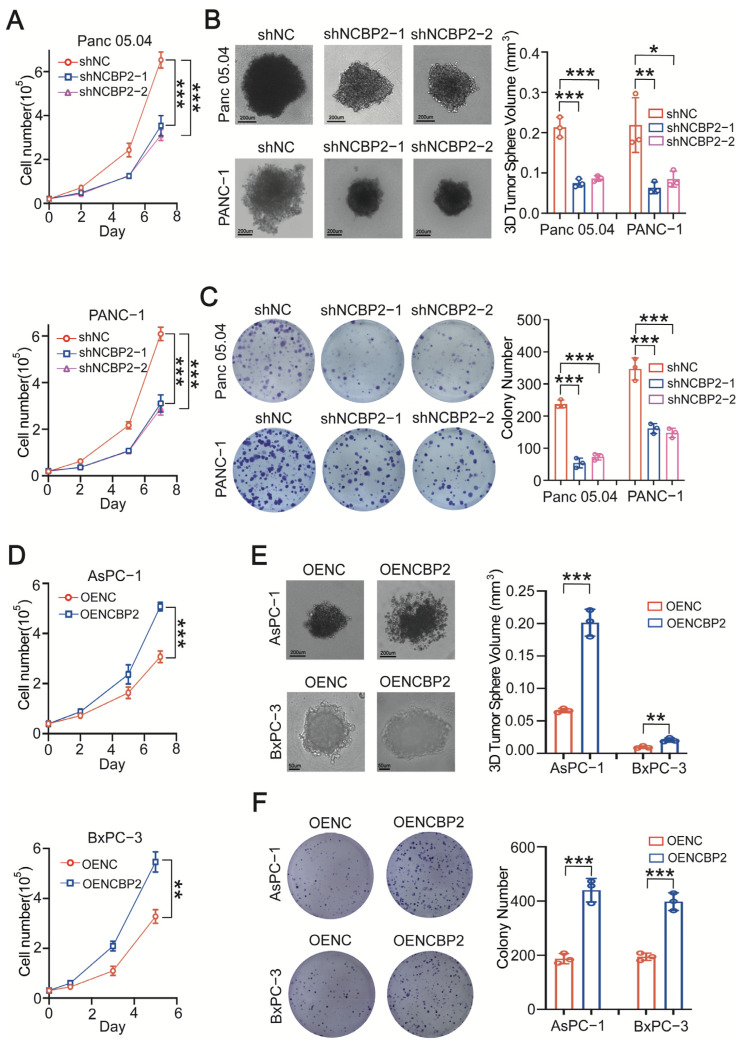
NCBP2 promotes PDAC cell growth in vitro. (**A**–**C**). The cell counting, 3D sphere, and colony formation assay results in control and NCBP2-knockdown Panc 05.04 and PANC-1 cells. (**D**–**F**). The cell counting, 3D sphere and colony formation assay results in control and NCBP2-overexpression AsPC-1 and BxPC-3 cells, * *p* < 0.05, ** *p* < 0.01, and *** *p* < 0.001.

**Figure 4 cancers-15-05454-f004:**
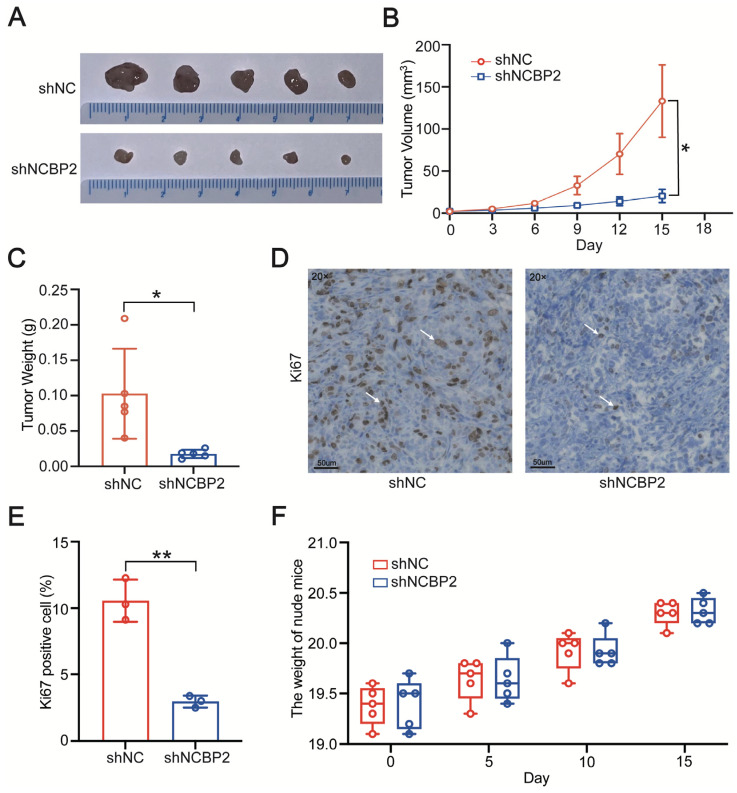
NCBP2 promotes PDAC cell growth in-vivo. (**A**,**B**) Tumor growth curves of xenograft models established from control or stable NCBP2-knockdown Panc 05.04 cells. (**C**) Assessment of tumor weight from control and NCBP2-knockdown groups. (**D**,**E**) The expression level of Ki-67 in tumor tissues from control and NCBP2-knockdown groups. (**F**) Body weight of nude mice in control and NCBP2-knockdown group, * *p* < 0.05, and ** *p* < 0.01.

**Figure 5 cancers-15-05454-f005:**
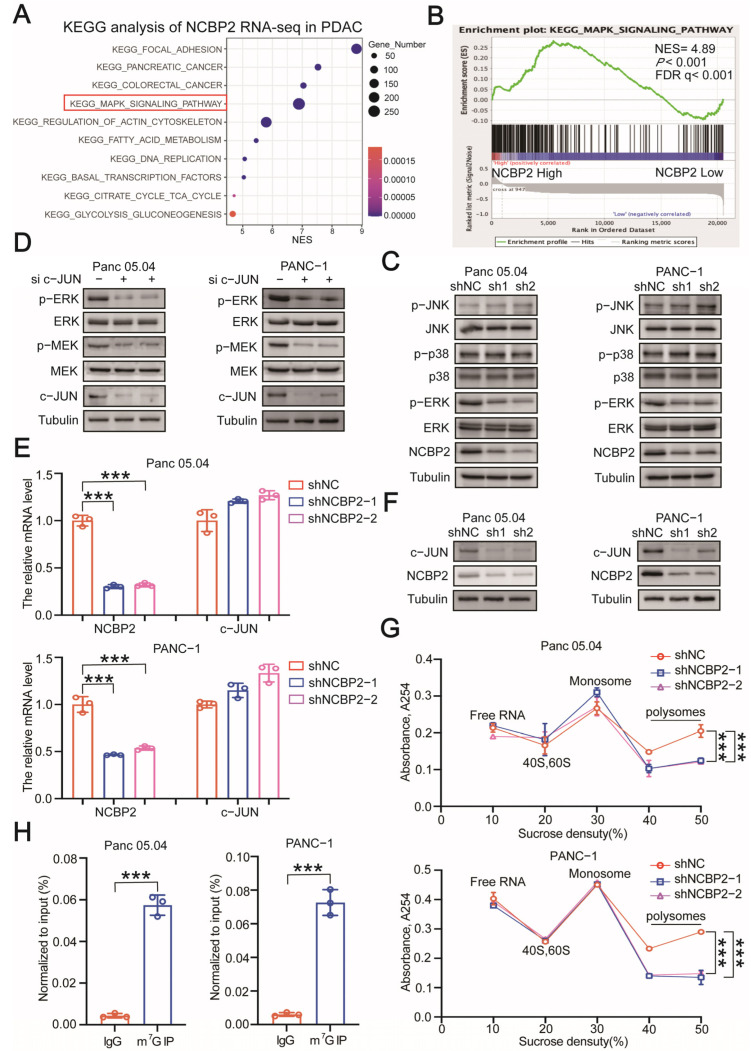
NCBP2 upregulates c-JUN to activate MEK/ERK signaling in a m^7^G-dependent manner. (**A**) KEGG analysis of RNA-seq data in PDAC patients with high or low NCBP2 expression from TCGA cohort. (**B**) The GSEA enrichment plot of “MAPK signaling pathway” in PDAC patients with high or low NCBP2 expression from TCGA cohort. (**C**) Immunoblotting for protein levels of total JNK/phosphorylated JNK (Thr183/Tyr185), total p38/phosphorylated p38 (Thr180/Tyr182), and total ERK/phosphorylated ERK (Thr202/Tyr204) in control and NCBP2-knockdown PDAC cells. Tubulin was used as the internal control. (**D**) Immunoblotting for protein levels of total/phosphorylated MEK and total/phosphorylated ERK (Thr202/Tyr204) in control and c-JUN-knockdown Panc 05.04 and PANC-1 cells. Tubulin was used as the internal control. (**E**) The mRNA expression levels of NCBP2 and c-JUN in control and NCBP2-knockdown Panc 05.04 and PANC-1 cells. (**F**) Immunoblotting for protein levels of NCBP2 and c-JUN in control and NCBP2-knockdown Panc 05.04 and PANC-1 cells. Tubulin was used as the internal control. (**G**) Polysome profiling results of the control and NCBP2-knockdown Panc 05.04 and PANC-1 cells. (**H**) Gene-specific m^7^G qPCR results for the m^7^G methylation levels of c-JUN in PANC 05.04 and PANC-1 cells, *** *p* < 0.001.

**Figure 6 cancers-15-05454-f006:**
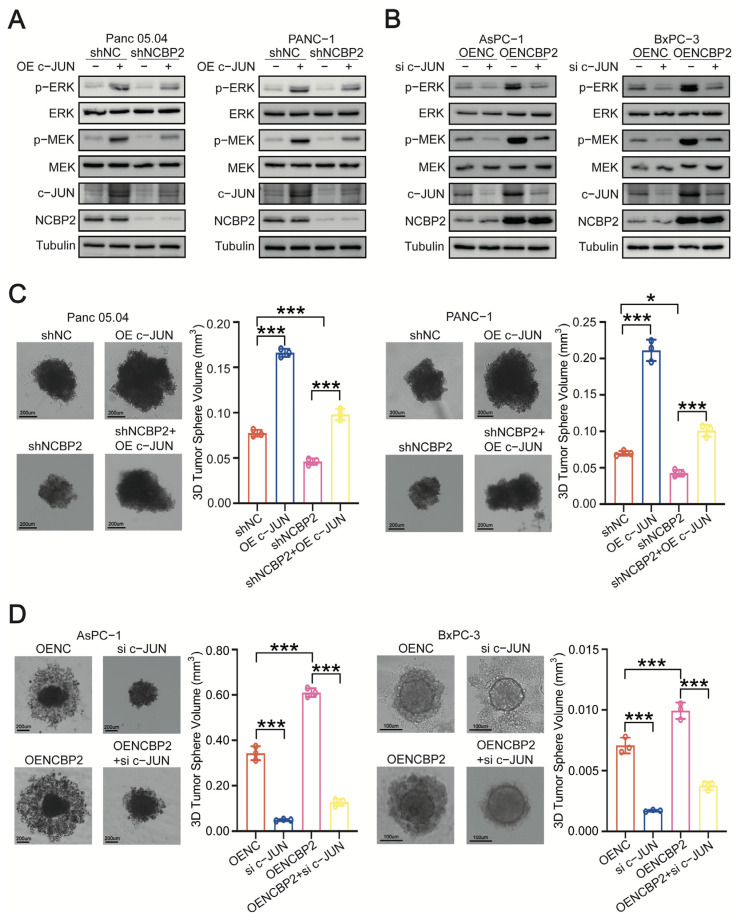
NCBP2 promotes PDAC progression by activating c-JUN/MEK/ERK signaling pathway. (**A**) Immunoblotting for protein levels of NCBP2, c-JUN, total/phosphorylated MEK, and total/phosphorylated ERK (Thr202/Tyr204) in control and NCBP2-knockdown PDAC cells overexpressing c-JUN. Tubulin was used as the internal control. (**B**) Immunoblotting for protein levels of NCBP2, c-JUN, total/phosphorylated MEK, and total/phosphorylated ERK (Thr202/Tyr204) in control and NCBP2-overexpression PDAC cells with c-JUN silenced. Tubulin was used as the internal control. (**C**) 3D sphere assays were performed after overexpressing c-JUN in control and NCBP2-knockdown PDAC cells. (**D**) 3D sphere assays were performed after c-JUN-knockdown in control and NCBP2-overexpressed PDAC cells, * *p* < 0.05 and *** *p* < 0.001.

**Figure 7 cancers-15-05454-f007:**
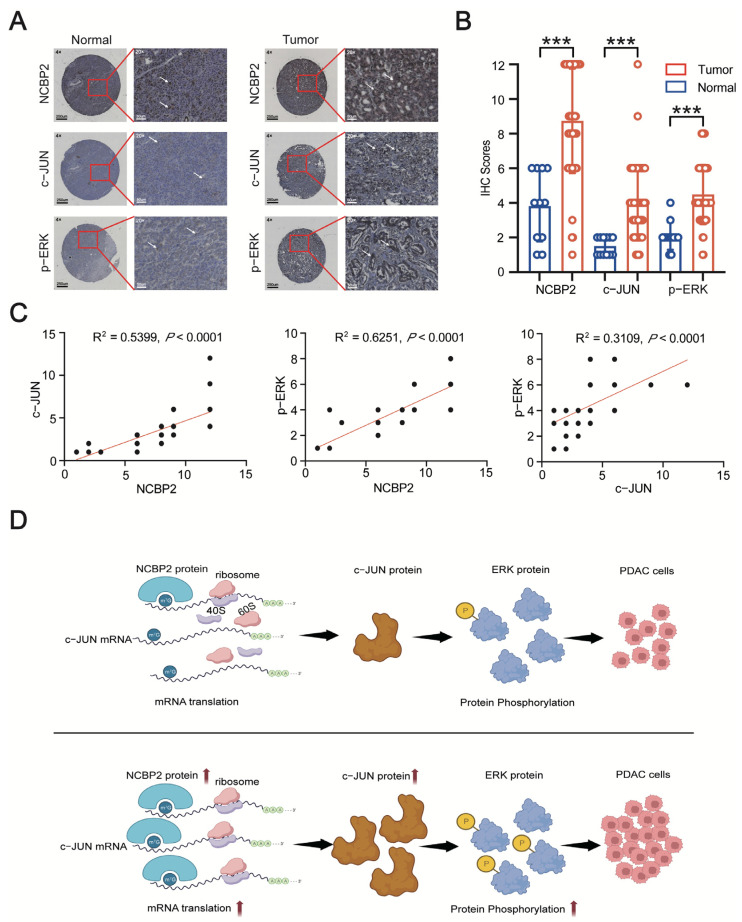
NCBP2 and c-JUN/MEK/ERK signaling pathways are significantly upregulated in PDAC patients. (**A**) The expression level of NCBP2, c-JUN, and p-ERK was determined by IHC analysis in tissue microarray. (**B**) The IHC scores of NCBP2, c-JUN, and p-ERK between PDAC and para-carcinoma tissues. (**C**) The correlation analysis results between NCBP2, c-JUN, and p-ERK based on their IHC scores in PDAC tissues. (**D**) A schematic model of NCBP2 function in PDAC. NCBP2 upregulates c-JUN-mediated MEK/ERK signaling activation in an m^7^G methylation-dependent manner, *** *p* < 0.001.

## Data Availability

The processed data that support the findings of this study are available from the corresponding author.

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
