# Peer review of "The m7G Reader NCBP2 Promotes Pancreatic Cancer Progression by Upregulating MAPK/ERK Signaling"

_cancers, 2023, doi:10.3390/cancers15225454_

Round 1

Reviewer 1 Report

Comments and Suggestions for Authors

Dear Authors,

This is a very interesting study showing the involvement of m7G methylation in PDAC development and progression. Using the gene knockdown and overexpression system in pancreatic cancer cell lines, the authors found that upregulation of NCBP2 activates MEK/ERK via activation of c-Jun translation and activates the growth of pancreatic cancer cells. Throughout the paper, the following revisions were felt necessary.

Major points:

1. The important thing is to clarify what is actually happening in the clinical specimen. Cell line results are only the results of cell lines, and must be verified in clinical specimens. Using clinical specimens, the NCBP2-c-Jun-MEK/ERK pathway should be verified by immunostaining and other methods. This will clarify how frequently this pathway is actually involved in pancreatic cancer. This will also be important for therapeutic strategies for pancreatic cancer.

2. In general, K-ras mutations are found in almost all pancreatic cancers and are thought to cause aberrant activation of the MAPK pathway. Please discuss how the activation of MEK/ERK by K-ras mutations and increased expression of NCBP2 are related.

3. GEPIA 2.0 uses data from TCGA and GTEx. The GEPIA 2.0 cohort that appears in Figure 2 is derived from the TCGA cohort (I do not know why there is one less example of low expression of NCBP2 in Figures 2E and 2F than in Figures 2C and 2D). Thus, Figures 2C and 2E and Figures 2D and 2F show the same pattern based on exactly the same sample. Delete either Figures 2C and D or Figures 2E and F. The text will also be modified accordingly.

Minor points:

1. Considering the function of NCBP2, which may be essential for cell proliferation and survival, what phenotype does knockout of NCBP2 exhibit? If the authors have any information, please describe it, as it will give us a clue as to whether suppression of NCBP2 expression can have a significant impact on normal cells when considering it as a therapeutic target for pancreatic cancer.

2. Line 99: GEO datasets are not RNA-seq data but DNA microarray data

3. Line 106-109: I don't think HPDE cells grow in DMEM or RPMI 1640 medium. I think the culture medium may be different.

4. Lines 136, 138, and 139: Please list the sources of pSin-vector, pLKO.1-puro vector, package plasmid pMD2.G and psPAX2.

5. Line 196: What are the 29 m7G-related genes analyzed?

6. Line 229: After "cell counting assay", insert (Figure 3A).

7. Line 231: Change Figure 3A-C to Figure 3B-C.

8. Line 241-243: When was the Ki-67-immunostained tissue from, and please describe in the text, such as Materials and Methods, or in the figure legend.

9. Line 248: GSEA, not GESA.

10. Line 382-383: "Although MEK inhibitor......KRAS mutant cancer patients [67-70]."is not a sentence.

11. All Figures, Supplementary Figures: What *, **, and *** indicate should be explained in each figure.

12. Figure 1: “Genes exclude” in two places is a mistake for “genes excluded”.

13. Figure 5A: The red square encloses "colorectal cancer". The authors might have meant to enclose "MAPK signaling pathway".

14. Table S3: Please add the dilution factor of the antibodies.

Comments on the Quality of English Language

The entire text was very easy to read; please carefully correct small errors like Minor point #9, 10.

Reviewer 2 Report

Comments and Suggestions for Authors

Pancreatic adenocarcinoma (PDAC) is a poor prognosis cancer, and therefore, much effort is currently invested to develop efficient drugs to cure it. Although much progress has been achieved recently in Ras inhibition that may be beneficial in up to 95% of the PDAC cases that are transformed by Ras mutants, not all mutants are currently targeted. Even in the case of the drug-sensitive mutants, there is a fast development of resistance with a very poor prognosis. Therefore, other strategies to combat the disease should be developed. Here the authors used bioinformatic approach and identified NCBP2 that regulates N7-methylguanosine (m7G) methylation and thereby also protein translation as a protein that may regulate PDAC survival.  They showed that knockdown of NCBP2 results in a reduced proliferation or organoid growth in cells expressing high levels of the protein. On the other hand, overexpression of NCBP2 induced higher cell growth. They also present some evidence that the effects are mediated by the translation of cJUN protein and the activation of ERK. 

            Overall, this is an interesting study that may lead to a desired new strategy to combat PDAC. The results are mostly clean and believable, although it is not clear how does cJUN induce ERK phosphorylation, and whether ERK is the component that signals to survival in the examined cells. These points as well as a few others listed below should be addressed in order to make this manuscript suitable for publication in Cancers. 

Comments:

1.     In figure 3, it is not explained well enough how knockdown of NCBP2 leads to the significant elevation of proliferation/survival of AsPC-1 and BxPC-3 cells that have only a minute expression of this protein. This point has to be explained also for Fig. 6.

2.     In figure 5, the authors follow ERK phosphorylation in “resting cells” which is assumed to be downstream of the mutated Ras upstream this cascade. However, the phosphorylation in these cells is generated by many regulators (phosphatases, scaffolds, etc) that are not necessarily related to the oncogenic signals. 

3.     Moreover, it is clear that cJun cannot induce ERK phosphorylation by itself, and is likely causing it by altering one of the regulators. This point has to be better discussed. It is very likely that the effect is mediated by MKPs, and therefore, it is suggested to show whether the reduced phosphorylation by CNBP2 knockdown can be reversed by MKP inhibitors. A hint regarding the regulator can be achieved also by showing whether the effect on ERK is all over the cell or just nuclear. 

4.     In continuation to point 2 and 3, and to eliminate the SteadyState effect of phosphatases, it is suggested to examine whether NCBP2 affect the phosphorylation of ERK shortly after stimulation (e.g. EGF).

5.     Another point that should be strengthen is to show whether the elevated phosphorylation by overexpressed CNBP2 or cJUN is indeed mediated by ERK. This can be done using ERK or MEK inhibitors. 

6.     Fig. S3A is confusing. Is the correlation presented for pMEK (like ERK) or MEK expression? What is MEK-tpm

7.     The authors state that Western blot analysis indicated that knockdown of NCBP2 had no effect on the stability of c-JUN protein in both Panc 05.04 and PANC-1 cells (Figure S3B). However, the results do show reduction in cJun expression. Please explain. In addition, it would be interesting to show whether there is any effect on cJUN phosphorylation (that can be mediated by all MAPKs including ERK).

8.     The Article is usually well-written However, the introduction is confusing and not very informative. Paragraph 2 lacks information and is not clear at all. The third paragraph is based on a very old review, and contains abbreviations that re not currently used (e.g. MKKK should be MAPKKK or MAP3K). There are 4 MAPK cascades, (ERK3/4 are not considered as a genuine cascade) and the components are not “family members”. It is suggested to consult with an expert during revision. 

9.     Throw-out the article, The Y axis should be specified (give information on what was measured)

Comments on the Quality of English Language

The English Language is fine. Only some minor points should be corrected

Reviewer 3 Report

Comments and Suggestions for Authors

In this study, entitled “The m7G reader NCBP2 promotes pancreatic cancer progression 2 by up-regulating MAPK/ERK signaling”, authors aimed to explore the role of NCBP2, a regulatory protein involved in m7G methylation, in PDAC progression. By using GEPIA and GEO database analyses, authors found that NCBP2 was highly expressed in tumors and associated with lower overall survival in PDAC patients. Their findings also showed that knockdown of NCBP2 diminished tumorigenicity of PDAC cells and reduced MEK/ERK activation in PDAC cells. Furthermore, the results indicate that c-JUN is a downstream target gene of NCBP2, and NCBP2 activates MEK/ERK signaling via regulating its translation efficiency in an m7G methylation-dependent manner. Overall, this study extends our knowledge to understand the possible mechanistic act of NCBP2 on promoting PDAC progression. The methods are clear and replicable, and the details of materials used in this study are fully described. The experiments have been properly conducted, and their conclusions are fully supported by the results. In fig.7, it is only suggested that authors may further address whether m7G methylation levels in cJUN mRNA differ between tumor and normal tissue (or between invasive and non-invasive PDAC cells).

Reviewer 4 Report

Comments and Suggestions for Authors

The introduction provides a clear explanation for the study by highlighting the lack of effective treatments for PDAC and the need to identify novel diagnostic and therapeutic targets. However, it would be beneficial to provide more context on the current challenges in PDAC research, such as the limited efficacy of existing therapies.

The key findings of the study, include the high expression of NCBP2 in PDAC patients, its correlation with poor prognosis, and its role in promoting PDAC cell proliferation. It also provides a concise explanation of the mechanistic link between NCBP2, c-JUN, and the MEK/ERK pathway.

Strengths:

1. The study demonstrates a comprehensive approach by integrating gene expression data and clinical information from multiple cohorts, enhancing the reliability of the findings.

2. The use of both knockdown and overexpression experiments adds credibility to the functional relevance of NCBP2 in PDAC cell proliferation.

3. The mechanistic insights into NCBP2's action shed light on the underlying pathways involved in PDAC progression.

Limitations & Suggestions for Improvement:

1. While the integration of data from TCGA and GEO cohorts is commendable, it would be beneficial to discuss potential limitations related to cohort heterogeneity, sample size variations, and potential biases associated with data collection and analysis. To address the limitations related to cohort heterogeneity, it would be valuable to perform subgroup analyses considering important clinical factors, such as tumor stage and patient demographics. This would provide a more nuanced understanding of the relationship between NCBP2 expression and PDAC prognosis.

2. The study primarily focuses on in vitro and in vivo experiments, which provide useful mechanistic insights. However, the lack of clinical validation limits the generalizability of the findings to PDAC patients.

3. The molecular mechanisms underlying NCBP2-mediated enhancement of c-JUN translation are briefly mentioned but not thoroughly investigated. A more detailed analysis would have strengthened the study's conclusions. For a comprehensive understanding of NCBP2-mediated c-JUN translation, the authors should provide more detailed experiments, such as ribosome profiling or polysome fractionation, to investigate the specific mechanisms involved and identify potential intermediary factors or regulators.

4. To overcome the lack of clinical validation, the authors should consider conducting further studies using patient-derived samples to confirm their findings and evaluate the potential prognostic or predictive value of NCBP2 expression in PDAC.

Overall, this study provides valuable insights into the role of NCBP2 in PDAC progression and highlights its potential as a therapeutic target. However, addressing the limitations and suggestions for improvement would strengthen the study's conclusions and their applicability to clinical settings.

Here are some detailed questions:

1. Inline 98/99/100, In this study, RNA-seq and related clinical data were obtained from TCGA-PDAC 98 and GEO datasets (GSE15471, GSE28735, GSE16515). Gene set enrichment analysis (GSEA) 99 software was used for gene enrichment analysis. The URLs of all online analysis websites 100 were shown in Table S1. In result one, a nice chart flow can be seen. Is there any possibility that this can be double-confirmed in human tissue chips or any clinical samples not just from a database?

2.     In figure 5A, line 289, KEGG analysis of RNA-seq data in PDAC patients with high or low NCBP2 expression 289 from TCGA-PDAC cohort. But in the figure, I see that [KEGG-COLORECTAL CANCER] was highlighted. What do you want to show in this figure section?

3.     In Figure 5, there is a general statement about the mechanistic role of NCBP2 in promoting PDAC progression through the hyperactivation of the RAF/MEK/ERK signaling pathway but do not elaborate on specific mechanisms or provide supporting evidence. Additional experiments or validation studies are necessary to confirm the role of NCBP2 in PDAC progression.

Comments on the Quality of English Language

No major issues with the English language.

Round 2

Reviewer 1 Report

Comments and Suggestions for Authors

Dear Authors,

Thanks for the revised manuscript and cover letter. I read it again throughout.

Thank you for your sincere and appropriate response to reviewers' comments. The addition of immunostaining data on clinical specimens, which several reviewers have pointed out, makes the authors' experimental data on cell lines more convincing and makes the paper more complete.

I believe the following points need to be corrected.

The resolution of Figures is generally low and appears blurry, so I would like to see the resolution increased so that they can be seen clearly. For example, due to poor resolution, it is not possible to evaluate the results of immunostaining in Figures 2G and 7A. Please increase resolution and indicate positive cells with arrows, etc. Also, for immunostaining of tissue arrays, please include the magnification of the left and right pictures. Furthermore, the position of P<0.05 at the bottom of Figure 1, between "The overall survival analysis in GEPIA 2.0-PDAC cohort (n=178)" and "NCBP2 " would be appropriate.

Comments on the Quality of English Language

I would like to see the revised English text proofread again by a native English speaker. I think several sentences are clearly incorrect. For example, P5 and P6. lines 192~199 have the protocol sentences as they are written and need to be corrected to normal sentences. Also, P6. lines 220~222 are not sentences.

Reviewer 2 Report

Comments and Suggestions for Authors

The authors successfully addressed my concerns.

I have no further comments 
